# More Nutritional Support on the Wards after a Previous Intensive Care Unit Stay: A nutritionDay Analysis in 136,667 Patients

**DOI:** 10.3390/nu15163545

**Published:** 2023-08-11

**Authors:** Arabella Fischer, Cecilia Veraar, Isabella Worf, Silvia Tarantino, Noemi Kiss, Christian Schuh, Michael Hiesmayr

**Affiliations:** 1Department of Anaesthesia, Intensive Care Medicine and Pain Medicine, Division of Cardiac, Thoracic, Vascular Anaesthesia and Intensive Care, Medical University of Vienna, 1090 Vienna, Austria; cecilia.veraar@meduniwien.ac.at; 2Center for Medical Data Science (CeDAS), Medical University of Vienna, 1090 Vienna, Austria; isabella.worf@meduniwien.ac.at (I.W.); silvia.tarantino@meduniwien.ac.at (S.T.); michael.hiesmayr@meduniwien.ac.at (M.H.); 3nutritionDay Worldwide, 1090 Vienna, Austria; 4Department of Health Economics, Center for Public Health, Medical University of Vienna, 1090 Vienna, Austria; noemikiss@live.com; 5IT-Systems and Communications (ITSC), Medical University of Vienna, 1090 Vienna, Austria; christian.schuh@meduniwien.ac.at

**Keywords:** nutrition, post-ICU, ICU, intensive care unit, hospitalisation, nutritionDay

## Abstract

ICU (intensive care unit) patients are exposed to nutritional risks such as swallowing problems and delayed gastric emptying. A previous ICU stay may affect nutritional support upon transfer to the ward. The aim was to study the use of enteral (EN), parenteral nutrition (PN), and oral nutritional supplements (ONS) in ward patients with and without a previous ICU stay, also referred to as post- and non-ICU patients. In total, 136,667 adult patients from the nutritionDay audit 2010–2019 were included. A previous ICU stay was defined as an ICU stay during the current hospitalisation before nutritionDay. About 10% of all patients were post-ICU patients. Post-ICU patients were more frequently exposed to risk factors such as a BMI < 18.5 kg/m^2^, weight loss, decreased mobility, fair or poor health status, less eating and a longer hospital length of stay before nDay. Two main results were shown. First, both post- and non-ICU patients were inadequately fed: About two thirds of patients eating less than half a meal did not receive EN, PN, or ONS. Second, post-ICU patients had a 1.3 to 2.0 higher chance to receive EN, PN, or ONS compared to non-ICU patients in multivariable models, accounting for sex, age, BMI, weight change, mobility, health status, amount eaten on nutritionDay, hospital length of stay, and surgical status. Based on these results, two future goals are suggested to improve nutritional support on the ward: first, insufficient eating should trigger nutritional therapy in both post- and non-ICU patients; second, medical caregivers should not neglect nutritional support in non-ICU patients.

## 1. Introduction

Insufficient nutritional therapy is a major concern in hospitalised patients [1]. Malnutrition is associated with a higher hospital length of stay (LOS) [2] and mortality [1,3]. Malnutrition should be screened and assessed in order to start adequate nutritional support [4,5]. The Global Leadership Initiative on Malnutrition (GLIM) diagnostic criteria for malnutrition are weight loss, low BMI, reduced muscle mass, reduced food intake, and acute or chronic disease [4]. Particularly, post-ICU (intensive care unit) patients are a vulnerable group because they suffer from long-term sequelae up to 8 years after ICU or hospital discharge [6,7,8,9].

On the one hand, transfer from the ICU to the ward may sensitise medical staff to watch for malnutrition in this vulnerable patient population. Post-ICU patients often experience swallowing problems after prolonged intubation or tracheostomy, lack of appetite, early satiety, delayed gastric emptying, and reduced gut motility [10,11,12]. On the other hand, transfer from the ICU to the ward may cause disruption of the pre-established nutritional therapy [12]. Upon transfer, noncommunication of any nutritional plan or the subsequent removal of the nasogastric tube on the ward may lead to insufficient nutritional therapy [12]. Standard hospital food alone can often not cover the energy requirements in post-ICU patients: hospital food provided alone covered merely 37 (21–67)% of predicted energy requirements [13]. In another study, post-ICU patients on hospital food alone received only 55–75% of their prescribed calories [14]. In total, 62% of post-ICU patients ate less than two thirds of their meal [15]. When hospital food was combined with ONS and EN, predicted energy requirements could be attained [13]. In sum, the inadequacy of nutritional support in post-ICU patients was often emphasised in previous studies, but nutritional support has never been compared between post- and non-ICU patients. It is, therefore, unknown whether and to what extent a previous ICU stay affects eating and nutritional therapy in the ward. Hence, the aim of the study was twofold: first, the association between eating and nutritional therapy was assessed in post- vs. non-ICU patients; second, the use of enteral (EN), parenteral nutrition (PN), and oral nutritional supplements (ONS) was assessed in post- vs. non-ICU patients while accounting for different risk factors in both groups. 

## 2. Materials and Methods

### 2.1. Study Design and Population

In short, nutritionDay is a worldwide annual audit, where participating wards prospectively collect data from all consenting patients present in the ward on a given day, usually a Thursday in November. This cross-sectional data collection represents the day-by-day patient population in a ward and the nutritional therapeutic interventions. All 136,667 adult patients from the nutritionDay audit from 2010 to 2019 were included. 

### 2.2. Outcome

The outcome was the use of any EN, PN, and ONS in patients on the ward with and without a previous ICU stay, also referred to as post- and non-ICU patients. The association between the use of any EN, PN, and ONS and eating was also studied. Eating referred to eating on nutritionDay. A previous ICU stay was defined as an ICU stay during the current hospitalisation before nutritionDay. Sex, age, BMI, weight change, mobility, self-related health status, eating on nutritionDay, hospital length of stay at nutritionDay, and surgical status were additional risk factors to account for in the multivariable model. Missing categories were used for each risk factor. The nutritionDay hospital questionnaires can be found under this link: https://www.nutritionday.org/en/-35-.languages/hospitals/english-metric-measures.html (accessed on 2 August 2023). Weight and height were answered by medical staff. Weight and height were either estimated (in 32% of patients during nutritionDay audit from 2016–2019) or measured (in 68% of patients during nutritionDay audit from 2016–2019). Weight loss was answered by the patient. The eating amount on nutritionDay was answered by the patient.

### 2.3. Statistical Analysis 

Demographic characteristics were described as count with percentage, median with interquartile range (IQR), and mean with standard deviation (SD), as appropriate. The comparisons of demographic characteristics and risk factors between non- and post-ICU patients were determined by calculating the odds of a previous ICU stay for a given risk factor in a univariable logistic model. For each risk factor, the reference group was the most frequent category or the category, containing the median of the population. Individual wards were considered as clusters. Next to the exposure of interest (previous ICU stay), all other risk factors associated with the use of EN, PN, or ONS were accounted for in the final logistic multivariable models. The interactions between the exposure of interest (previous ICU stay) and all risk factors were considered. Graphs display odds ratios with 95% confidence intervals. Due to multiple comparisons, the significance level was set at 0.01. All analyses were performed with STATA 15.1. Graphs were performed with PRISM 9.

## 3. Results

### 3.1. Patient Population

A total of 136,667 patients from 6952 wards in 67 countries were included. Thirty-nine percent of all patients were surgical patients. About half of the patients were female. The median age was 66 [52; 78] years, and BMI was 24.8 [21.7; 28.7] kg/m^2^. Forty-four percent of patients experienced weight loss during the previous 3 months. Fourteen percent of patients ate nothing on nutritionDay. Ten percent of all patients had a previous ICU stay and were identified as post-ICU patients (Table 1). The median hospital LOS before nutritionDay was 6 [2; 14] days. The observed hospital mortality was 3.5%. 

### 3.2. Risk Factors in Post- and Non-ICU Patients

Men were more frequent in post- than non-ICU patients. An age younger than 30 and older than 80 years was less frequent in post- than non-ICU patients (Table 2). 

In general, post-ICU patients were exposed to more risk factors than non-ICU patients: a BMI < 18.5 kg/m^2^ was more frequent, whereas a BMI ≥ 30 kg/m^2^ was less frequent in post-ICU patients. Weight loss or unknown weight change was more frequent in post-ICU patients. Decreased mobility needing assistance or a bedridden state were more frequent in post- than non-ICU patients. A very good health status was less frequent in post-ICU patients. Post-ICU patients ate less frequently the entire meal. Post-ICU patients had a longer length of stay before nutritionDay. There were more patients without surgery in the non-ICU than the post-ICU group (63.2 vs. 40.8%) (Table 2). 

### 3.3. Association between Eating and Nutritional Therapy in Post- vs. Non-ICU Patients

Both the post- and non-ICU patients were inadequately fed: in patients eating nothing on nutritionDay, 71 and 44% of the non- and post-ICU patients did not receive any nutritional therapy, respectively. EN was only prescribed in 8 and 22% of the non- and post-ICU patients, respectively, PN in only 10 and 15% of the non- and post-ICU patients, respectively, and ONS in only 8% of the non- and post-ICU patients eating nothing, respectively. In patients eating less than half a meal, 79 and 68% of the non- and post-ICU patients did not receive any nutritional therapy, respectively. The most frequent nutritional therapy was ONS, prescribed in 13 and 17% of the non- and post-ICU patients. Less frequently, EN was prescribed in 3 and 6% of the non- and post-ICU patients and PN in 2 and 4% of the non- and post-ICU patients eating less than half a meal (Figure 1 and Appendix A). In general, the patients receiving EN or PN had a higher 30-day hospital mortality than patients without EN or PN. The patients eating nothing or eating less than half a meal had a higher 30-day hospital mortality than those eating a full meal (Appendix A).

### 3.4. Use of EN, PN, and ONS in Post- vs. Non-ICU Patients

Interestingly, the post-ICU patients had higher odds to receive EN [OR 1.98 (95% CI, 1.82–2.16)], PN [OR 1.50 (95% CI, 1.35–1.67)], and ONS [OR 1.34 (95% CI, 1.25–1.44)] compared to the non-ICU patients when accounting for all risk factors in the multivariable models (Figure 2 and Appendix A). 

In addition, men had higher odds to receive EN and PN. Younger patients below 40 years had lower odds to receive EN, PN, and ONS. Patients with a BMI > 25 kg/m^2^ had a lower chance to receive EN, PN, and ONS than underweight patients with a BMI < 18.5 kg/m^2^. The patients with weight loss had a higher chance to receive EN, PN, and ONS. Bedridden patients had higher odds to receive EN and ONS than the patients walking alone. The patients with a fair or poor health status had a higher chance to receive EN, PN, and ONS. The odds to receive EN were higher for the patients eating nothing. The odds to receive PN and ONS were higher for the patients eating less than half a meal to nothing. The longer the length of hospital stay, the higher were the odds to receive EN, PN, or ONS. Surgical patients had a lower chance to receive EN and ONS but a higher chance to receive PN compared to the medical patients (Figure 2 and Appendix A). 

## 4. Discussion

For the first time, we showed to what extent a previous ICU stay affects eating and nutritional therapy in the ward. First, both post- and non-ICU patients were inadequately fed: only about one third of the patients eating less than half a meal received nutritional therapy, and, second, the post-ICU patients had a 1.3 to 2.0 higher chance to receive EN, PN, or ONS compared to the non-ICU patients in multivariable models accounting for sex, age, BMI, weight change, mobility, health status, amount eaten on nutritionDay, hospital length of stay, and surgical status.

### 4.1. Risk Factors in Post- and Non-ICU Patients

In comparison to non-ICU patients, post-ICU patients were unsurprisingly more frequently exposed to risk factors such as a BMI < 18.5 kg/m^2^, weight loss, decreased mobility, fair or poor health status, less eating, and a longer hospital LOS before nutritionDay.

### 4.2. Association between Eating and Nutritional Therapy in Post- vs. Non-ICU Patients

All the post- and non-ICU patients were inadequately fed: only about one third of patients eating less than half a meal received nutritional therapy. Apparently, insufficient eating does not always trigger nutritional support. Improving nutritional support in patients with insufficient eating is crucial, as patients eating a quarter of a meal are known to have a two times higher mortality risk [1].

### 4.3. Use of EN, PN, and ONS in Post- vs. Non-ICU Patients

The post-ICU patients had a 1.3 to 2.0 higher chance to receive EN, PN, or ONS than non-ICU patients in multivariable models accounting for all risk factors. A previous ICU stay may, therefore, trigger the use of nutritional support. Medical staff may be sensitised to the nutritional risks in post-ICU patients, such as swallowing problems, early satiety, delayed gastric emptying, and reduced gut motility [10,11,12]. Moreover, post-ICU patients already have gastric tubes and central venous lines in place, which may encourage use of EN or PN. However, discontinuity in nutritional therapy was reported upon transfer from the ICU to the ward [12]. Discontinuity was due to the noncommunication of any nutritional plan or the subsequent removal of any nasogastric tube in the ward [12]. Overall, the pre-establishment of any nutritional treatment, including lines and tubes during a previous ICU stay, still seems to promote use of nutritional support in comparison to the non-ICU patients in our analysis. The inadequacy of nutritional support in post-ICU patients was often emphasised in previous studies [13,14,15]. This was verified by our data, but the inadequacy of nutritional support was even worse in non-ICU patients. We showed this for the first time, as no previous study compared post- to non-ICU patients.

In addition, the GLIM diagnostic criteria for malnutrition [4], namely, weight loss, low BMI, eating less than half of the meal, and reduced health status as a surrogate for disease, increased the odds to receive EN, PN, or ONS in the multivariable analyses. Moreover, decreased mobility and a longer hospital LOS increased the odds to receive EN, PN, or ONS. Mobility status is not part of the GLIM diagnostic criteria for malnutrition [4], but it is clearly related to the use of nutritional support. A bedridden patient is most often not able or motivated to eat the entire meal. The longer the patient stays in the hospital, the more the medical staff may think about prescribing nutritional support. Surgical pre- and postoperative patients had high odds to receive PN but lower odds to receive ONS. ONS was not often prescribed to preoperative patients, according to our analysis. Yet, preoperative ONS is strongly recommended according to the Enhanced Recovery After Surgery (ERAS) guidelines because ONS ameliorates general well-being, decreased protein breakdown, and postoperative insulin resistance [17]. The required preoperative fasting state may explain why preoperative patients did not receive EN. Elderly patients above 80 years had higher odds to receive ONS than younger patients below 40 years. Higher age may be an additional trigger to prescribe ONS, but younger patients in need for ONS should not be disregarded in clinical practice.

### 4.4. Strength and Limitations

The strength of our analysis was the real-world study design of nutritionDay in nearly 140,000 patients from both surgical and medical wards of the entire world.

One limitation of our analysis is the unavailability of ICU data in the post-ICU patient group: the nutritionDay project on the ward focuses on nutrition and clinical data on the ward. Therefore, we did not have any information about ICU length of stay or ICU disease severity. However, we presented the available data of hospital length of stay before nutritionDay, which included ICU length of stay. Of note patients staying longer in the hospital were more likely to be included in a cross-sectional study, like the nutritionDay project. However, a cross-sectional design depicts the daily activities of a hospital. Moreover, large databases tend to determine risk factors with very little clinical relevance in spite of a large univariate effect. Therefore, we presented a multivariable model, where invalid associations are less likely.

## 5. Conclusions

Only about one third of all the patients eating less than half a meal received EN, PN, or ONS. Interestingly, the post-ICU patients had a 1.3 to 2.0 higher chance to receive EN, PN, or ONS than the non-ICU patients after accounting for all risk factors. Therefore, two future goals are suggested to improve nutritional support on the ward: first, insufficient eating should trigger nutritional therapy in both post- and non-ICU patients, and, second, medical caregivers should not neglect nutritional support in non-ICU patients.

## Figures and Tables

**Figure 1 nutrients-15-03545-f001:**
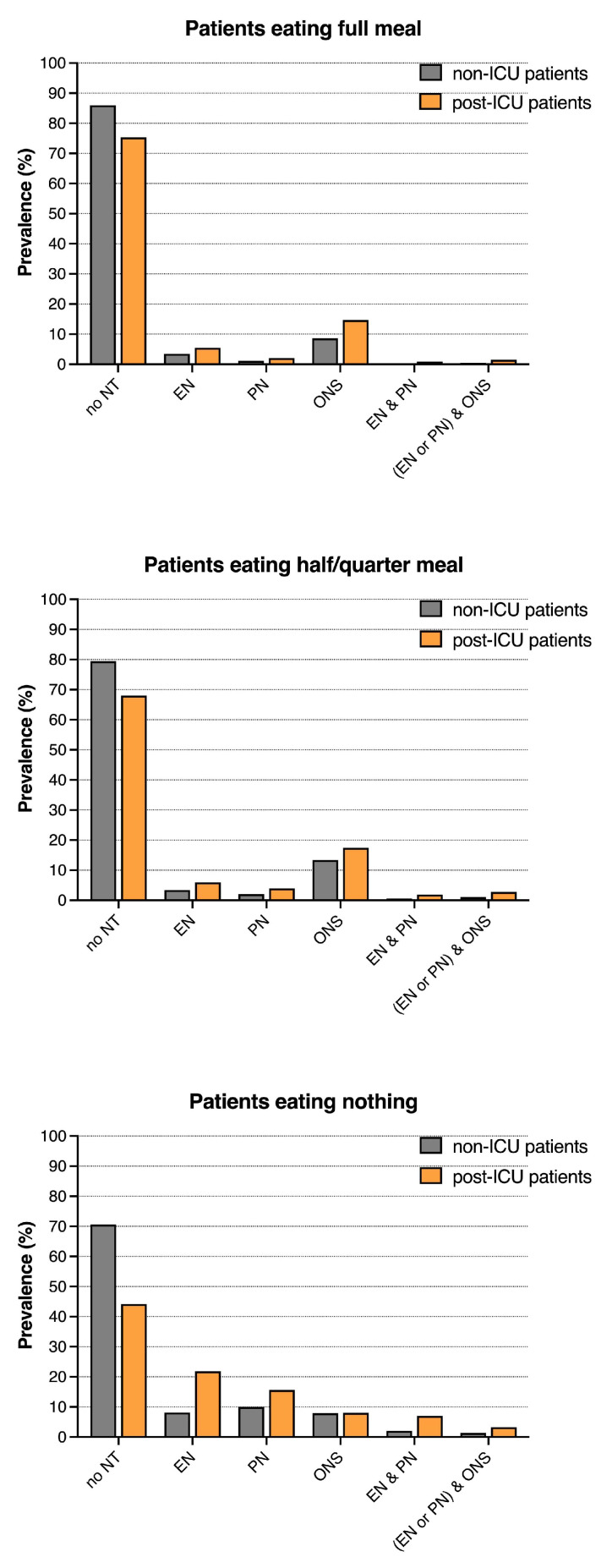
Nutritional therapy in patients eating the full meal, half to a quarter or nothing on nutritionDay, n = 136,667 (Missing data were excluded). NT = nutritional therapy, EN = enteral nutrition, PN = parenteral nutrition, and ONS = oral nutritional supplements.

**Figure 2 nutrients-15-03545-f002:**
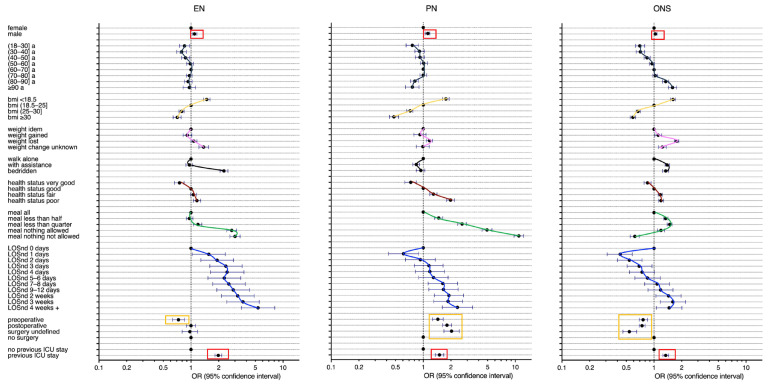
Use of any enteral nutrition (EN), parenteral nutrition (PN), or oral nutritional supplement (ONS) associated with a previous ICU stay and other risk factors; multivariable logistic regression, n = 136,667 (Missing data were excluded). The pattern of each risk factor was traced in colour. LOSnd: Hospital length of stay before nutritionDay, a: years, BMI: in kg/m^2^.

**Table 1 nutrients-15-03545-t001:** Demographic characteristics, n = 136,667.

	Median [Q1; Q3] or Mean ± SD or n (%)
Age, years	66 [52; 78]
Sex (female/male/unknown), %	49.7%/49.5%/0.8%
BMI, kg/m^2^	24.8 [21.7; 28.7]
BMI > 30, n (%)	24,086 (17.6%)
BMI < 18.5, n (%)	9790 (7.2%)
Weight loss during previous 3 months, n (%)	60,356 (44.2%)
Full meal eaten on nutritionDay, n (%)	54,619 (40.0%)
Nothing eaten on nutritionDay, n (%)	19,298 (14.1%)
Enteral nutrition, n (%)	9751 (7.1%)
Parenteral nutrition, n (%)	6552 (4.8%)
Oral nutritional supplements, n (%)	16,186 (11.8%)
Previous ICU stay, n (%)	14,293 (10.5%)
Previous ICU stay in postoperative patients, n postop patients with previous ICU stay/n all postop patients (%)	7052/35,475 (19.9%)
Surgical patients, n (%)	53,484 (39.1%)
Hospital LOS before nutritionDay, days	6 [2; 14]
PANDORA score, points [16]	26 [19; 33]
Predicted hospital mortality from PANDORA score, %	3.4 ± 4.9%
Observed 30-day hospital mortality, n (%)	4205/121,346 (3.5%)

LOS: Length of stay.

**Table 2 nutrients-15-03545-t002:** Demographic characteristics and risk factors in non- and post-ICU patients before nutritionDay (nDay), n = 136,667.

Risk Factor	Category	All Patients		Non-ICU Patients		Post-ICU Patients	
		n	%	n	%	n	%
		136,667		122,374		14,293	
Sex	female (reference)	67,922	(49.7%)	61,609	(50.3%)	6313	(44.2%)
	male ***	67,705	(49.5%)	59,833	(48.9%)	7872	(55.1%)
	sex missing	1040	(0.8%)	932	(0.8%)	108	(0.8%)
Age	(18–30] a ***	7948	(5.8%)	7206	(5.9%)	742	(5.2%)
	(30–40] a ***	9006	(6.6%)	8150	(6.7%)	856	(6.0%)
	(40–50] a ***	12,469	(9.1%)	11,129	(9.1%)	1340	(9.4%)
	(50–60] a	20,079	(14.7%)	17,749	(14.5%)	2330	(16.3%)
	(60–70] a (reference)	27,597	(20.2%)	24,287	(19.8%)	3310	(23.2%)
	(70–80] a **	28,665	(21.0%)	25,443	(20.8%)	3222	(22.5%)
	(80–90] a ***	23,873	(17.5%)	21,802	(17.8%)	2071	(14.5%)
	≥90 a ***	6590	(4.8%)	6182	(5.1%)	408	(2.9%)
	age missing ***	440	(0.3%)	426	(0.3%)	14	(0.1%)
BMI	<18.5 *	9790	(7.2%)	8632	(7.1%)	1158	(8.1%)
	(18.5–25] (reference)	53,986	(39.5%)	48,098	(39.3%)	5888	(41.2%)
	(25–30] **	35,981	(26.3%)	32,299	(26.4%)	3682	(25.8%)
	≥30 ***	24,086	(17.6%)	21,699	(17.7%)	2387	(16.7%)
	BMI missing **	12,824	(9.4%)	11,646	(9.5%)	1178	(8.2%)
Weight	weight idem (reference)	45,952	(33.6%)	41,869	(34.2%)	4083	(28.6%)
change	weight gained ***	11,824	(8.7%)	10,910	(8.9%)	914	(6.4%)
	weight lost ***	60,356	(44.2%)	53,064	(43.4%)	7292	(51.0%)
	weight change unknown ***	10,053	(7.4%)	8887	(7.3%)	1166	(8.2%)
	weight change missing	8482	(6.2%)	7644	(6.2%)	838	(5.9%)
Mobility	walk alone (reference)	77,713	(56.9%)	71,139	(58.1%)	6574	(46.0%)
	with assistance ***	33,985	(24.9%)	29,877	(24.4%)	4108	(28.7%)
	bedridden ***	14,772	(10.8%)	12,234	(10.0%)	2538	(17.8%)
	mobility missing ***	10,197	(7.5%)	9124	(7.5%)	1073	(7.5%)
Health	very good ***	13,185	(9.6%)	12,061	(9.9%)	1124	(7.9%)
status	good (reference)	42,254	(30.9%)	37,953	(31.0%)	4301	(30.1%)
	fair **	46,891	(34.3%)	41,770	(34.1%)	5121	(35.8%)
	poor **	24,626	(18.0%)	21,900	(17.9%)	2726	(19.1%)
	health status missing	9711	(7.1%)	8690	(7.1%)	1021	(7.1%)
Eating on nDay	all (reference)	54,619	(40.0%)	49,679	(40.6%)	4940	(34.6%)
	less than half ***	33,186	(24.3%)	29,657	(24.2%)	3529	(24.7%)
	less than quarter ***	18,446	(13.5%)	16,301	(13.3%)	2145	(15.0%)
	nothing allowed ***	8342	(6.1%)	7298	(6.0%)	1044	(7.3%)
	nothing not allowed ***	10,956	(8.0%)	9712	(7.9%)	1244	(8.7%)
	meal missing ***	11,118	(8.1%)	9727	(7.9%)	1391	(9.7%)
Hospital length of stay before nDay (days)	0 (reference)	1815	(1.3%)	1736	(1.4%)	79	(0.6%)
	1 **	15,063	(11.0%)	14,667	(12.0%)	396	(2.8%)
	2	16,776	(12.3%)	16,127	(13.2%)	649	(4.5%)
	3	14,113	(10.3%)	13,312	(10.9%)	801	(5.6%)
	4 ***	10,729	(7.9%)	9879	(8.1%)	850	(5.9%)
	5–6 ***	12,708	(9.3%)	11,503	(9.4%)	1205	(8.4%)
	7–8 ***	13,595	(9.9%)	12,035	(9.8%)	1560	(10.9%)
	9–12 ***	14,106	(10.3%)	12,174	(9.9%)	1932	(13.5%)
	2 weeks ***	11,971	(8.8%)	10,119	(8.3%)	1852	(13.0%)
	3 weeks ***	12,502	(9.1%)	10,132	(8.3%)	2370	(16.6%)
	4 weeks+ ***	9264	(6.8%)	7124	(5.8%)	2140	(15.0%)
	duration before missing ***	4025	(2.9%)	3566	(2.9%)	459	(3.2%)
Surgical status	preoperative	10,236	(7.5%)	9459	(7.7%)	777	(5.4%)
	postoperative ***	35,475	(26.0%)	28,423	(23.2%)	7052	(49.3%)
	surgery undefined time	7773	(5.7%)	7146	(5.8%)	627	(4.4%)
	no surgery (reference)	83,183	(60.9%)	77,346	(63.2%)	5837	(40.8%)

*** *p* < 0.0001, ** *p* < 0.001, and * *p* < 0.01, a: years, BMI in kg/m^2^.

## Data Availability

Datasets can be accessed upon request.

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
