# Peer review of "More Nutritional Support on the Wards after a Previous Intensive Care Unit Stay: A nutritionDay Analysis in 136,667 Patients"

_nutrients, 2023, doi:10.3390/nu15163545_

Round 1
Reviewer 1 Report
In my opinion, this study emphasizes a very interesting aspect of the nutritional management of hospitalized patients. It is well known that malnutrition is common in hospitalized acute patients, but this work provides specific knowledge about the state of post ICU patients and about their nutritional treatment that reveal a high degree of insufficiency. The study is useful to raise awareness about the need for strategies to improve the nutritional management of post-ICU patients.
The message is clear and relevant, but there are some concerns about how the authors present the results. The authors present separate data in two cohorts because the questionnaire was slightly modified in 2016. However, the authors do not specify what the changes were in the questionnaire to understand the need to present two cohorts. Despite the fact that there are statistically significant differences between the groups, they are generally not clinically relevant and reading cohort 1 or cohort 2 separately will not modify the conclusions, so what is the point of presenting both cohorts separately? In my opinion, it only makes the results more difficult to read, so I would like to suggest presenting the entire cohort in a single analysis.
Another concern is the mischaracterization of post-ICU patients. Could you present more information about this study population (APACHE II, ICU length of stay,...): If not possible, it should be described as a limitation.
Is nutritional management different between medical and surgical patients? Can you comment on that?
Finally, and the most important. Nutritional therapy of patients who eat nothing, less than a quarter or less than half is described in Table S1. I think it is very important to separate the patients who in this situation receive TPN, EN or ONS, and to highlight that there is a large percentage of patients who, despite receiving less than half the meal, do not receive any complementary nutritional treatment. Did they have a worse prognosis, longer hospital stay, or mortality? Perhaps a figure and including this topic in the discussion could send a very important message in this question. In my opinion, one of the most important messages of the work is underfeeding associated with oral nutrition in hospitalized patients.
On the other hand, the multivariate analyzes presented in figures 2, 3 and 4 do not impose any relevant message and could be presented as supplementary material.
Reviewer 2 Report
Thank you for submitting a nutrition support study
Introduction
The energy deficit issue is highlighted with no mention of protein targets or delivery.
Line 51 – define ‘actual’ eating compared to eating. Do the authors mean recorded intakes?
Line 51 – nutritional therapy ‘at’ the ward should be rephrased to either ‘at the ward level’ or ‘on the ward’.
Methods
The methods section could do with expansion of detail. Population information is very brief.
The readers would need more information on the ICU patients than ‘an ICU stay’. How sick were the patients? What was their length of stay on ICU? Were they mechanically ventilated or not? An ICU patient with a short stay is very different to a longer stay, mechanically ventilated multi-organ failure patient. And patients in the latter group display significantly higher rates of muscle wasting (Puthucheary et al, 2013).
How was the weight and weight change determined? Was this actual weight or estimated weight? If estimated, how? How was height assessed? There are many different methods.
How was ‘eating on nutritionDay’ assessed? By whom? Were dietitians involved?
Results
There was a very good number of patients involved in this study
Paragraph 3.2, lines 101- 109: none of this data in terms of risk factors is new.
Par. 3.3, nothing here is new to the literature, this is already known. That post ICU patients require more nutrition therapy, whether it be EN, ON or ONS, is known due to their level of debilitation and the insults they experienced during the ICU phase of illness. Although some patients may have ‘pre-established nutritional treatment’ it should also be acknowledged that NGT for instance are often removed once patients are extubated.
If patients don’t eat, the odds of receiving EN is higher. Post-operative patients receive more ONS/EN, again none of this is new data.
Discussion
As per results, this data does not add anything new to the literature. All hospital patients are inadequately fed, and ICU patients more than non ICU patients. Elderly patients receive more ONS than those below 40 years of age, no new data either.
A point of interest is however in line 155. ‘Only about one third of patients eating less than half of a meal received nutritional therapy’. That point is worth exploring and discussing. What are the reasons behind this? This will be something worth exploring so institutions can improve in this area.
Also, what are the fasting practices pre-op, this will obviously vary between institutions but why are patients not on ERAS?
Strengths have been listed, but limitations not.
In my view the following are limitations to this study and manuscipt:
· A one day snapshot audit has some value but the limitations thereof should also be mentioned.
· This is retrospective data from 2010-2019.
· Detail is lacking in methods to describe the ICU cohort
· ‘Multivariate model where invalid associations are less likely’, but they can still occur.
· The results do not add to the current literature, apart from the one point highlighted above, line 155.
Conclusion
Line 214 – change ‘did get EN’ to ‘received EN’.
Many thanks
No major issue with English, a few minor suggestions have been added above.
Reviewer 3 Report
The aim of this paper was to study the use of enteral, parenteral nutrition and oral nutritional supplements in ward patients with and without a previous ICU stay, also referred to as post- and non-ICU patients. Post-ICU patients had a 1,3 to 2,5 higher chance to get EN, PN or ONS compared to non-ICU patients in multivariable models accounting for sex, age, BMI, weight change, mobility, health status, amount eaten on nutritionDay, hospital length of stay and surgical status, suggesting that medical caregivers at the ward should enhance nutritional therapy
not only in post-ICU patients, but also in non-ICU patients.
The study is well executed in its design, analysis of results and conclusions. There is a great need for this type of study to promote the interest of adequate nutritional support in these vulnerable patients.
It would be interesting to complete the introduction and justification of the article with a comprehensive review that includes more related bibliographic references.
